# Evaluation of Temperature Sensors for Detection of Heat Sources Using Additive Printing Method

**DOI:** 10.3390/s22218308

**Published:** 2022-10-29

**Authors:** Ju-Hun Ahn, Han-Na Kim, Jin Yeon Cho, Jeong Ho Kim, Chang-Yull Lee

**Affiliations:** Department of Aerospace Engineering, Inha University, Incheon 22212, Korea

**Keywords:** EHD ink-jet printing, temperature, resistance, printing sensor, heat source

## Abstract

Electrohydrodynamic (EHD) inkjet printing is an efficient technique for printing multiple sensors in a multifaceted area. It can be applied to various fields according to the shape of the printing result and the algorithm employed. In this study, temperature sensors capable of detecting heat sources were fabricated. Inks suitable for EHD inkjet printing were produced, and optimal parameters for printing were determined. Printing was performed using the corresponding parameters, and various printing results were obtained. Furthermore, an experiment was conducted to confirm the temperature measurement characteristics of the results and the tolerance of the sensor. Grid-type sensors were fabricated based on the results, and the sensor characteristics were confirmed in an orthogonal form. Heat was applied to arbitrary positions. Resistance to changes due to heat was measured, and the location at which the heat was generated was detected by varying the change in resistance. Through this study, efficient heat control can be achieved, as the location of the heat source can be identified quickly.

## 1. Introduction

Satellites are especially most sensitive to heat as they are operated in an extreme environment, which directly affects the electronic equipment inside the satellite. Errors in the electronic equipment inside the satellite can cause mission failure. In particular, since a satellite cannot be repaired when errors occur, thermal analysis and thermal design of the satellite are essential [1]. In addition, the use of temperature measurement technology is a necessary for the thermal control of satellites [2]. However, since commercialized temperature sensors occupy a part of the space of satellites, research on manufacturing temperature sensors for satellites is in progress [3]. The technology to measure heat is playing a very important role in the field of satellites. Temperature sensors that measure temperature are divided into contact type and non-contact type according to the measurement method. Among these, the contact method is primarily used for accurate measurements. However, the commercially available contact temperature sensor has a disadvantage in that it occupies a certain space inside the satellite and is not freely attached due to its limited shape. In addition, it is difficult to use a technology that detects heat sources. To overcome this limitation, Mulyadi et al. studied thin-film temperature sensors using barium strontium titanate [4]. Liu et al. fabricated flexible temperature sensors using polyethyleneimine and reduced graphene oxide. These temperature sensors can be used for real-time measurements [5]. In these studies, it was possible to measure curved surfaces; however, detection of the location of heat generation was not considered. It was considered that the heat-source detection method could be solved using printed electronics.

Printed electronics technology has high potential for various applications. These technologies are particularly suitable for manufacturing flexible sensors and displays. Printed electronic technologies can be manufactured in desired shapes and sizes. In addition, it has low material consumption, which is an advantage of the printing technology. Various types of printed electronics technologies have been developed [6]. Contact printing and non-contact printing were used according to the printing method. Screen printing and flexography were used for contact printing. Screen printing is simple and can print on curved surfaces by pushing inks onto the screen; however, the print quality is relatively poor [7]. Flexography is a printing method that uses rollers and is mainly used for printing antennas and organic electronic circuits [8,9]. It has the advantages of high resolution and fast printing speed; however, it is difficult to print curved surfaces. Noncontact printing includes aerosol and inkjet printing. Compared to contact printing, non-contact printing does not require a printing plate. Aerosol printing is a technology that jets inks at high speeds using compressed air or ultrasonic waves [10]. There is no problem with nozzle clogging, and it has a high resolution; however, it cannot be used for mass production.

Ink-jet printing is a technology in which ink is dropped because the force of gravity. It can print with high resolution using a simple method. Among the various printed electronic technologies, inkjet printing is well known as the most successful approach [11]. EHD inkjet printing is an improved inkjet printing technology. While inkjet printing technology can only obtain results of several hundred micrometers or more because of the surface tension of the inks, EHD inkjet printing uses electrical force to break down the surface tension. Thus, printing results of a few micrometers or less can be obtained [12]. Sensor fabrication using inkjet printing has been studied extensively. Ando et al. developed a mass sensor by taking advantage of the low cost of the printing technology. This mass sensor has excellent resolution [13]. The most studied sensors used in printing technology are gas and strain sensors. Alshammari et al. fabricated a gas sensor by applying carbon nanotubes to inkjet printing [14]. Rehberger et al. conducted research on the fabrication of strain sensors using inkjet printing. The strain sensors used silver nanoinks and printed various patterns with different characteristics [15]. Ink-jet printing research on the fabrication of temperature sensors has also been reported. Dankoco et al. experimented by printing a thermistor material on polyimide substrates [16]. Choi et al. fabricated a flexible temperature sensor using EHD inkjet printing. These temperature sensors used drop-on-demand, which is an EHD inkjet printing ejection mode, and they have thermistor characteristics that measure temperature by changing the resistance [17]. Therefore, printing technology is suitable for manufacturing various sensors. However, there are only a few studies focusing on temperature sensors compared to those of gas and strain sensors. In addition, although research on flexible temperature sensors has progressed, studies on determining the location of heat generation are still insufficient. Therefore, studies were conducted on a method for finding the location of heat generation through a sensor manufactured by printing technology.

In this study, temperature sensors are fabricated using EHD inkjet printing, an improved printed electronics technology. Inks that can be used for EHD inkjet printing technology are manufactured and applied. Resistance measurements are performed to confirm the characteristics of the temperature sensors based on the number of repeated prints. In addition, the sensing characteristics are derived by measuring the resistances that changed according to temperature. The tolerance of the sensor is confirmed by performing temperature cycling, and the results show the most satisfactory measurement characteristics. Experiments are conducted to detect a heat source by arranging the temperature sensors obtained through the experiments in an orthogonal shape. According to these experiments, temperature sensors can be manufactured using EHD inkjet printing. Moreover, experiments confirmed the possibility of fabricating a sensor to detect a heat source, which is the location where heat is generated.

## 2. EHD Ink-Jet Printing

### 2.1. Prepared Inks

Figure 1 shows the production mechanism and the amount of material added to the inks, which have been used in the fabrication of temperature sensors. The solvent was mixed with α-terpineol (≥95%, Kanto Chemical Co., Inc., Tokyo, Japan) and diethylene glycol monobutyl ether acetate (DGBA; ≥98%, Daejung, Korea) for approximately 2 h using a stirrer [18]. A ceramic powder with a Curie point that changed the temperature according to a specific temperature was added to the prepared solvent. The ceramic was dispersed in a solvent for approximately 4 h. Finally, Ag paste (Changsung Nanotech Co., Korea) was added to achieve conductivity. The solution to which all the materials were added dispersed the particles for approximately 24 h at 900 RPM. The ink was prepared using this method.

### 2.2. Printing Mechanism

Figure 2a shows a schematic of the EHD inkjet printing system. EHD inkjet printing has several printing parameters. These parameters were controlled using the hardware and software. The configuration of the printing system contained conductive ink in a syringe, which was discharged through a nozzle. The discharge flow rates of these conductive inks were finely controlled using a flow controller connected to a syringe. The linewidth of the result is determined by the flow rate used for printing, and the flow rate conditions used depend on the viscosity characteristics of the ink. Subsequently, *x*-, *y*-, and *z*-axis controllers control the motor system. The motor determines the speed of printing. The faster the printing speed, the thinner the line width that can be obtained, but it may be difficult to obtain conductivity due to disconnection. Therefore, it is necessary to control the speed according to the usage method of the printing results. The part that makes EHD inkjet printing possible is the high-voltage controller. Figure 2b presents the principle of EHD inkjet printing using a high-voltage controller [19]. The voltage output from the high-voltage controller acted on the nozzle and substrate. The electric field breaks down the surface tension of the inks forming a circular meniscus. Inks with collapsed surface tension are ejected in the form of cone-jet, which is the most ideal meniscus form in EHD inkjet printing. The three controllers are operated using a computer, and the formed meniscus can be checked using a CCD camera.

### 2.3. Printing Applied

Experiments were conducted to apply the inks to EHD inkjet printing. EHD inkjet printing has different conditions depending on the manufacturing process and the characteristics of the inks. Therefore, the shape of the meniscus based on the voltage was confirmed. Figure 3 shows the meniscus shapes that change according to the voltage of the prepared inks. Experiments were carried out at a printing height of 1000 µm and a flow rate of 9 μL/min to confirm the meniscus. In addition, the voltage was increased from 0 to 3 kV by 0.1 kV, and the measurement was carried out. Meniscus shapes were classified as dripping, micro-dripping, cone jet, unstable cone jet, and multijet [20]. Dripping is the section where the surface tension of the inks is stronger than the applied voltage; therefore, they are ejected in the form of water droplets. Micro-dripping is the section where the meniscus shape starts to change because of the voltage. However, the surface tension has not yet been completely broken; thus, the shape is characterized by pulling the shape of a water droplet. The cone jet is the most suitable section for EHD inkjet printing since the surface tension of the ink is completely broken. The unstable cone jet was in the same state as the cone jet, and the surface tension was broken. However, the meniscus significantly vibrates because the force of the voltage is slightly stronger than the force of gravity. A multijet is a section in which overvoltage occurs because the voltage is very high compared to the resistance of the inks. The prepared ink formed a cone-jet at 1.9 to 2.2 kV under voltage conditions. Therefore, the relevant section was the most suitable parameter.

Printing was performed under a voltage of 2 kV, based on the results obtained through mechanism-confirmation experiments. The printing speed was 30 mm/s, and polyimide films with excellent thermal properties were used as printing substrates. Repeat printing was performed to determine the resistance and sensing characteristics according to the number of prints. The printing was repeated from steps 1 to 5, and each step represented the number of prints. The sensors that had completed printing were completed by thermal curing. The linewidths for each step were confirmed. Figure 4 shows the results of optical microscopy taken to confirm the line width, and Figure 4f shows a graph summarizing the measured line widths. Based on the measurement of the line width, it was the thinnest in step 1 (300 μm), and the thickest was obtained in step 5 (1000 μm). In addition, the linewidth increased by approximately 200 μm as the number of steps increased.

Subsequently, resistance measurements were performed for each step. Resistance measurement was performed with a multimeter through electrodes located at both ends, and the average was obtained by performing five times for each step. In step 5, which had the thickest line thickness, and step 4, 17.85 and 26.88 Ω were measured, respectively. Steps 3 and 2 measured 47.18 and 93.23 Ω, respectively. However, it was impossible to measure the resistance in step 1. It was confirmed that the resistance increased by approximately two times as the number of steps was lowered. As a result, continuous printing does not cause loss because the resistance is lowered by the amount of added material.

Scanning electron microscopy (SEM) imaging was performed to confirm the particle arrangement of the printed inks and the reason for their resistance characteristics. Figure 5 shows the SEM images for each step. The sufficiently small particles were Ag particles, and the larger particles were ceramic particles. In the case of ceramic particles, nanoscale powders were added. However, agglomeration occurred because the van der Waals force, which is the attraction force between the nanoparticles, significantly acted. This phenomenon causes the ceramic particles to become larger. According to the SEM imaging results, the resistance decreased as the level increased. This is because the higher the level, the more densely the particles are filled and the smoother the flow of electricity. In addition, there is a problem with the connection of particles because there is a lot of space between the particles printed in step 1. Therefore, resistance measurement is difficult in step 1.

## 3. Temperature Sensing

### 3.1. Measurement of Printing Sensors

Changes in the resistance of the manufactured temperature sensor were measured. Figure 6 shows a schematic of the resistance measurement according to the temperature change of the sensor. A high-temperature environment was used in the chamber, and the printed temperature sensor was placed at the center of the chamber. Electrodes were attached to both sides to measure the resistance change of the printed temperature sensor. Thermocouples were also used to measure the temperature inside the chamber. For the measurements, the attached electrode and thermocouple were simultaneously collected using the DAQ. The temperature was varied from room temperature (20 °C) to 180 °C.

Figure 7 shows the temperature measurements. In the first step, the resistance measurement was not possible. Therefore, the temperature test was impossible. Table 1 summarizes the sensitivity and temperature measurement range of the manufactured sensors. The sensitivity of the sensors was 0.075, 0.07, 0.065, and 0.06 Ω/°C in steps, and the sensitivity decreased as the stage increased. Conversely, the detection ranges were 160, 165, 170, and 180 °C, and the higher the steps, the wider the detection range. As a result of the experiments, it was confirmed that the stage in which the resistance is continuously increasing under the set temperature condition is step 5. Therefore, step 5 was selected as the final model for the experiments.

We performed temperature cycling tests to evaluate the tolerance of the sensor under the conditions selected from the temperature measurement results. Temperature cycling was performed for three cycles, and the temperature was measured up to 140 °C, which is the maximum usable temperature of the cycling tester. Figure 8 shows the results of the temperature cycling tests. The trend of the measured temperature data and the resistance data of the manufactured sensor proceeded similarly. Additionally, a peak point was formed at a similar position. These results confirmed that the tolerance of the sensor was stable.

### 3.2. Detecting of Heat Source

Experiments were conducted to detect the heat source using a printed sensor manufactured through the experiments. Figure 9 shows the temperature sensors fabricated for heat-source detection. The temperature sensors are composed of two layers, and the two layers are made to be orthogonal to each other. Therefore, the manufactured temperature sensor consists of *X*- and *Y*-axis layers. Ten lines are placed in each layer. Additionally, the resistance of each wire to change because of heat was measured.

Figure 10 shows the heat-source detection results. Heaters were arranged at arbitrary positions to track the heat source. The positions of the heaters are shown in Figure 10a. The measurement experiment was performed, while the heater and sensor were in contact. Electrodes were connected to each printed wire, and the resistance varying with temperature was measured. Figure 10b shows a visualization of the measured resistance data obtained using the manufactured temperature sensor. This was visualized using the difference value obtained by subtracting the resistance before heating from the resistance after heating. As a result of the measurement shown in Figure 10b, the *Y*-axis had the largest resistance difference at Y7 and Y8. Additionally, on the *X*-axis, there was a large difference in the resistance at X5, X6, X7, and X8. The points at which X and Y intersect can be estimated as the heat generation positions. Consequently, it coincides with the position of the heater, which is arbitrarily designated. Therefore, a printed sensor capable of tracking a heat source was manufactured.

## 4. Conclusions

Experiments were conducted to fabricate sensors that could detect the location of heat generation in a temperature sensor. First, inks suitable for EHD inkjet printing were prepared, and experiments were conducted to determine the optimal parameters by examining the meniscus shapes. Sections in which a meniscus of a stable shape was formed were derived. In addition, experiments were conducted to confirm that the resistance characteristics changed with repeated printing. The resistance characteristics improve as the number of printing cycles increases. Experiments were conducted to measure the change in resistance with temperature using the printing results of several steps. Printing results that could be continuously measured under the conditions set through the experimental results were selected. Temperature cycling experiments were conducted to check the stability of the sensor using the printing result with the best measurement performance. The temperature and resistance changes in the cycling experiments showed similar trends. Thus, sensors using EHD inkjet printing were successfully manufactured.

Heat source detection experiments were performed by attaching the manufactured sensors in an orthogonal form. The heaters were attached at random locations in a heat-source sensing environment. The attached heater was operated, and the resistance of the sensors changed because of the heat was measured. The difference between the two resistances was determined by measuring the resistance of the sensor before and after heating. The resistance difference was arranged according to the layers, and the position where the two resistance differences intersected was derived.

## Figures and Tables

**Figure 1 sensors-22-08308-f001:**
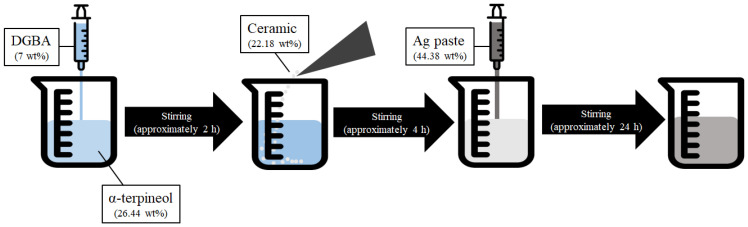
Fabrication mechanism of inks for EHD ink-jet printing.

**Figure 2 sensors-22-08308-f002:**
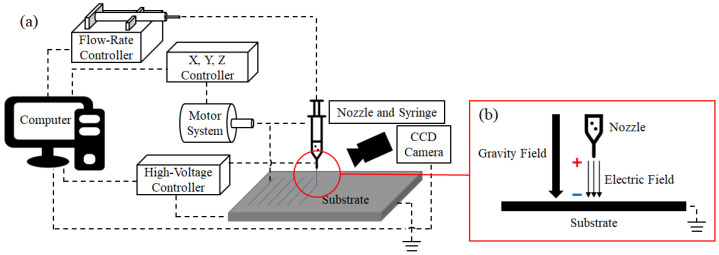
(**a**) System schematic diagram of EHD ink-jet printing. (**b**) Principle of EHD ink-jet printing.

**Figure 3 sensors-22-08308-f003:**
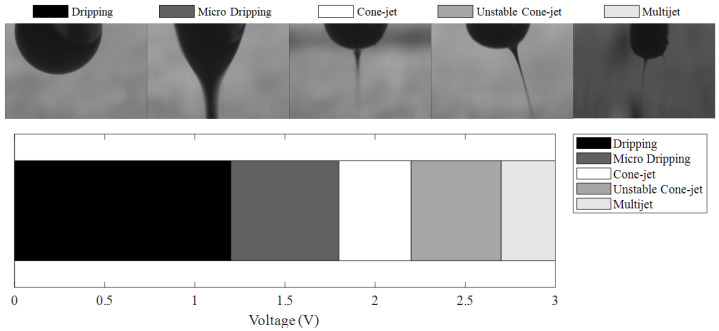
Diagram of meniscus stability for voltage conditions and meniscus shapes.

**Figure 4 sensors-22-08308-f004:**
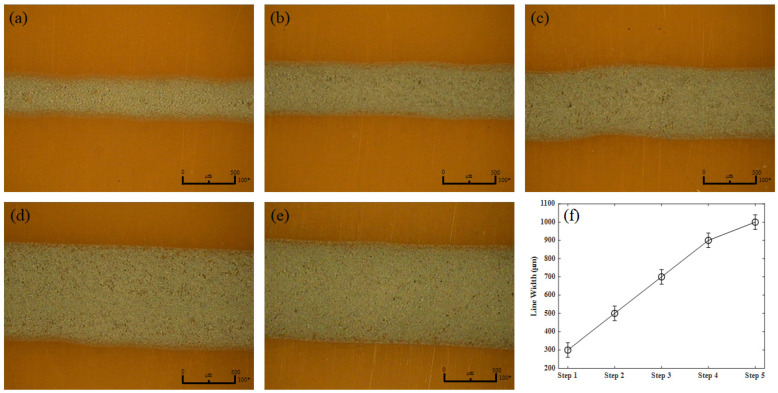
Line width imaging of the printing results using optical microscopes for steps (**a**) 1, (**b**) 2, (**c**) 3, (**d**) 4, and (**e**) 5. (**f**) Line width according to the step of the printing results.

**Figure 5 sensors-22-08308-f005:**
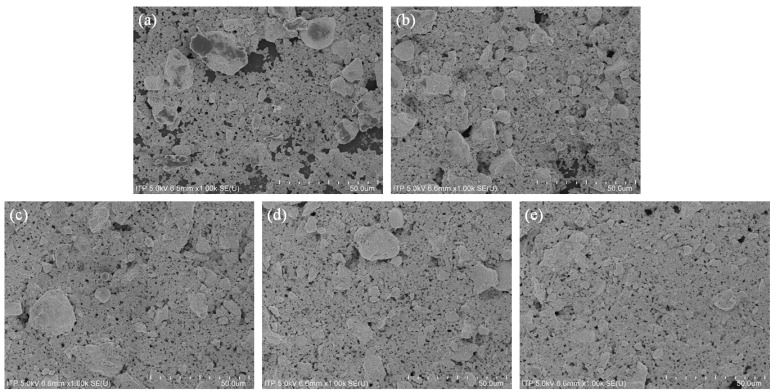
SEM results according to each step to confirm the arrangement of particles for steps (**a**) 1, (**b**) 2, (**c**) 3, (**d**) 4, and (**e**) 5.

**Figure 6 sensors-22-08308-f006:**
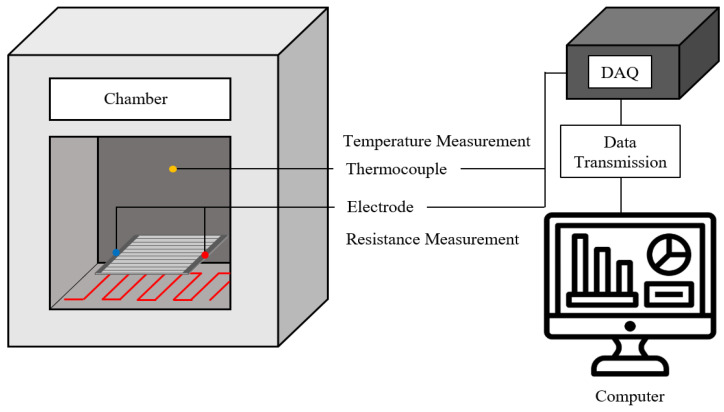
Schematic diagram of resistance measurement of fabricated temperature sensors.

**Figure 7 sensors-22-08308-f007:**
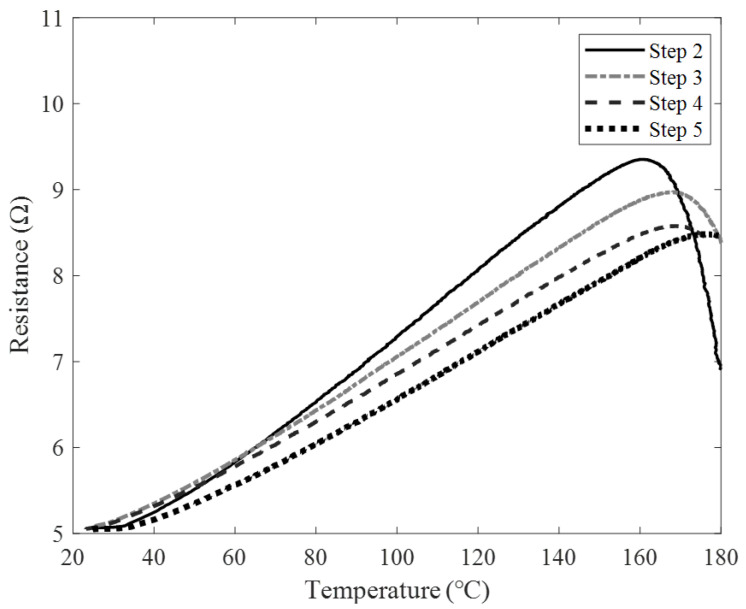
History of resistance change according to temperature change by steps.

**Figure 8 sensors-22-08308-f008:**
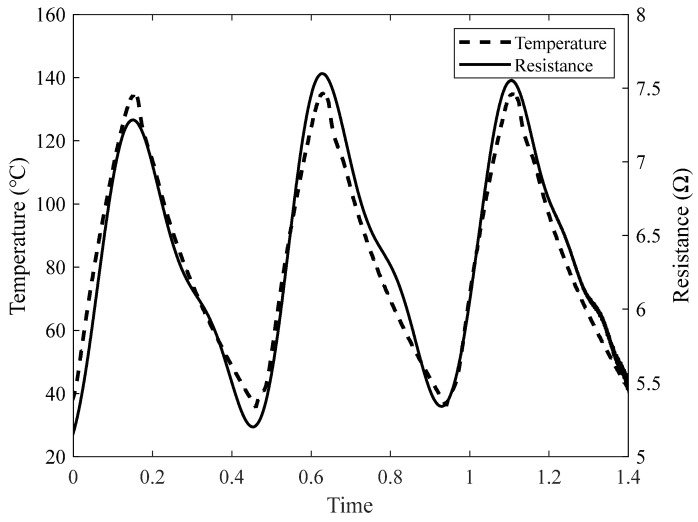
Temperature cycling results of the fabricated temperature sensor.

**Figure 9 sensors-22-08308-f009:**
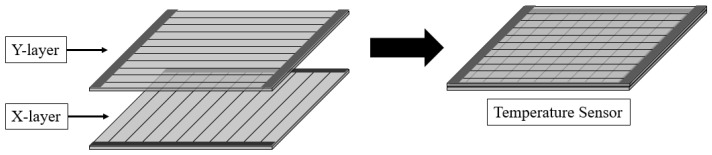
Temperature sensors modeling for heat source detection.

**Figure 10 sensors-22-08308-f010:**
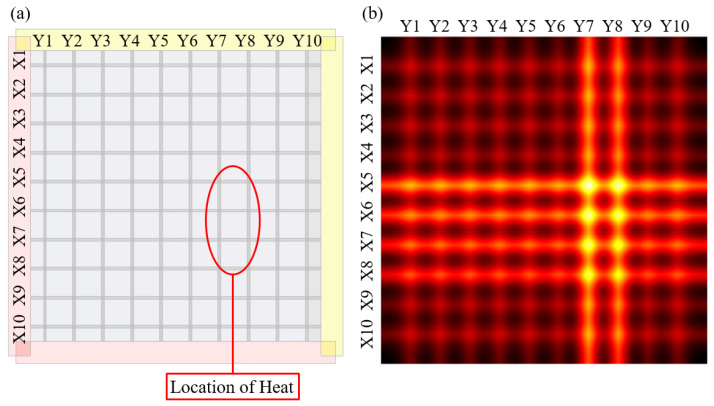
Heating position and visualized heat detecting technology.

**Table 1 sensors-22-08308-t001:** Sensitivity and temperature measurement range according to the steps of the manufactured sensors.

	Sensitivity (Ω/°C)	Temp. Range (°C)
Step 1	X	X
Step 2	0.075	160
Step 3	0.07	168
Step 4	0.065	170
Step 5	0.06	180

## Data Availability

The data are available upon request from the corresponding authors.

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
