# Peer review of "Evaluation of Temperature Sensors for Detection of Heat Sources Using Additive Printing Method"

_sensors, 2022, doi:10.3390/s22218308_

Round 1

Reviewer 1 Report

The manuscript reports the demonstration of a temperature sensor made by inkjet printing. The research could be interesting to researchers in related field. However, due to the reasons below, I think the manuscript cannot be considered for publication in its current form.

Many inkjet-printed temperature sensors have been demonstrated. The authors should elaborate more on the innovation and features of the sensor reported in this paper. In the current form, it is difficult to tell the significance of the research.

The manuscript provides a detailed description on the inkjet printing process. What is the difference between the process in this manuscript and the typical inkjet printing process?

The manuscript does not provide sufficient sensing data to show the performance of the sensor. For instance, what is the accuracy and range of the temperature sensing?  What is the spacial resolution of the grid-type sensor?

Reviewer 2 Report

I recommend its publication with minor revision and re-review as listed below.

1. What are the main applications of the prepared sensor and used method during the analysis ?

2. Amount of material added to the 101 inks, which have been used in the fabrication of temperature sensors- what type of materials used  

3. In the introduction, there is no information about the adverse effect of sensor. Authors are encouraged to add this information.

4. In the introduction part author should compare the applications of method over other methods and this should be actualized with the following literature

5. Experiments were conducted to apply the inks to EHD inkjet printing- specify the parameter of the ink

6. Figure 2(b) presents the 123 principle of EHD inkjet printing using a high-voltage controller- please justifies with proof.

7.  Resistance measurements were performed for each step- did this experiments with EIS OR ..

8.  Error bar missing

9. Purity of the chemicals missing

10. Author should recheck all the abbreviations, figures, and tables and their captions and calculated results.

11. What is the sensitivity of the prepared sensor?

12. Information related to the used method and materials is less in the introduction part, the author should improve with more information.

13. Need comparison table with some of the recent sensors.

Round 2

Reviewer 1 Report

The authors have addressed my concerns and I therefore recommend the manuscript to be considered for publication.